# DEEP CONTINUOUS NETWORKS

## ABSTRACT

CNNs and computational models of biological vision share some fundamental principles, which, combined with recent developments in deep learning, have opened up new avenues of research in neuroscience. However, in contrast to biological models, conventional CNN architectures are based on spatio-temporally discrete representations, and thus cannot accommodate certain aspects of biological complexity such as continuously varying receptive field sizes and temporal dynamics of neuronal responses. Here we propose deep continuous networks (DCNs), which combine spatially continuous convolutional filter representations, with the continuous time framework of neural ODEs. This allows us to learn the spatial support of the filters during training, as well as model the temporal evolution of feature maps, linking DCNs closely to biological models. We show that DCNs are versatile. Experimentally, we demonstrate their applicability to a standard classification problem, where they allow for parameter reductions and meta-parametrization. We illustrate the biological plausibility of the scale distributions learned by DCNs and explore their performance in a pattern completion task, which is inspired by models from computational neuroscience. Finally, we suggest that the continuous representations learned by DCNs may enable computationally efficient implementations.

## 1 INTRODUCTION

Computational neuroscience and computer vision have a long and mutually beneficial history of cross-pollination of ideas (Sejnowski, 2020; Cox & Dean, 2014). The current state-of-the-art in computer vision relies heavily on deep neural networks (DNNs), and in particular convolutional neural networks (CNNs), from which multiple analogies can be drawn to biological circuits (Kietzmann et al., 2018). Specifically, recent advances in DNNs have enabled researchers to learn more accurate models of the response properties of neurons in the visual cortex (Klindt et al., 2017; Cadena et al., 2019; Ecker et al., 2019), as well as to test decades old hypotheses from neuroscience in the domain of computer vision (Lindsey et al., 2019). However, contrary to biological models, CNNs typically operate in the domain of spatio-temporally discrete signals, and employ appropriately discretized kernels, as a natural part of digital image processing.

In computational neuroscience, on the other hand, large scale neural network models of the visual system often adopt continuous, closed-form expressions to describe spatio-temporal receptive fields, as well as the interaction strength between populations of neurons (Dayan & Abbott, 2001). Among others, such descriptions serve to limit the scope and parameter space of a model, by utilizing prior information regarding receptive field shapes (Jones & Palmer, 1987) and principles of perceptual grouping (Li, 1998). In addition, the choice of continuous—and often analytic—functions help retain some analytical tractability in complex models involving a large number of coupled populations. Our approach draws inspiration from such computational models to propose continuous representations of receptive fields in CNNs, where both the shape and the scale of the filters are trainable in the continuous domain.

In a complementary fashion, recent influential work in deep learning has introduced neural ordinary differential equations (ODEs) (Lu et al., 2018; Ruthotto & Haber, 2019; Chen et al., 2018) which propose a continuous time (or depth) interpretation of CNNs. Such continuous time models both offer end-to-end training capabilities with backpropagation which are highly applicable to computer vision problems (e.g. by way of adopting ResNet blocks (He et al., 2016)), as well as help bridge the gap to computational biology where networks are often modelled as dynamical systems which

evolve according to differential equations. In this work we aim to extend the impetus of the continuous time neural ODEs to the spatio-temporal domain.

To that end we introduce deep continuous networks (DCNs), which are spatio-temporally continuous in that the neurons have spatially well-defined receptive fields based on scale-spaces and Gaussian derivatives (Florack et al., 1996) and their activations evolve according to equations of motion comprising convolutional layers. We combine spatial and temporal continuity in a network with neural ODEs by learning linear weights for a set of analytic basis functions (as opposed to pixel-based weights), which can also intuitively be parametrized as a function of time, or network depth.

The following outlines our main contributions: (i) We provide a theoretical formulation of spatio-temporally continuous deep networks building on Gaussian derivative basis functions and neural ODEs; (ii) We demonstrate the applicability of DCN models, namely, that they exhibit a reduction in parameters, and can be used to parametrize convolutional filters as a function of time in a straight-forward fashion, while achieving performance comparable with or better than ResNet and ODE-Net baselines; (iii) We show that filter scales learned by DCNs are consistent with biological observations and we propose that the combination of our design choices for spatial and temporal continuity may be helpful in studying the emergence of biological receptive field properties as well as high-level phenomena such as pattern completion; (iv) We suggest that the continuous representations learned by DCNs may be leveraged for computational savings.

We believe DCNs can bring together two communities as they provide a test bed for hypotheses and predictions pertaining to both biological systems as well as pushing the boundaries of biologically inspired computer vision.

## 2 DEEP CONTINUOUS NETWORKS

### 2.1 NEUROSCIENTIFIC MOTIVATION

There is little doubt that modern deep learning frameworks will be conducive to effective and insightful collaborations between neuroscience and machine learning (Richards et al., 2019). In particular in vision research, CNNs are becoming increasingly popular for modelling early visual areas (Batty et al., 2017; Ecker et al., 2019; Lindsey et al., 2019). Here we propose a model which can facilitate such investigations by linking the end-to-end trainable but discrete CNN architectures with the biologically more plausible and spatio-temporally continuous computational models.

**Structured receptive fields.** Classical receptive fields (RFs) of cortical neurons display complex response properties with a wide array of selectivity structures already at early visual areas (Van den Bergh et al., 2010). Such response properties may also vary greatly based on multiple factors. For example the RF size (spatial extent) is known to depend on eccentricity (Harvey & Dumoulin, 2011) and visual area (Smith et al., 2001) and may even change with depth within a cortical layer (Bauer et al., 1999). Similarly, studies have shown that receptive field size and spatial frequency selectivity of neurons may co-vary with input contrast (Sceniak et al., 2002).

Based on these observations, we aim to build a model which can accommodate the biological realism better than conventional CNNs, by explicitly modelling the RF size as a trainable parameter. To that end, we adopt a Gaussian scale-space representation for the convolutional filters, which we call structured receptive fields (SRFs) (Jacobsen et al., 2016). Previously, Gaussian scale-spaces have been proposed as a plausible model of biological receptive fields and feature extraction in low-level vision (Florack et al., 1992; Lindeberg & Florack, 1994; Lindeberg, 1993). Here, we are inspired partially by computational models which investigate the origin of response properties in the visual system, by employing RFs and recurrent interaction functions which scale as a difference of Gaussians (Somers et al., 1995; Ernst et al., 2001). Partially, we are motivated by the success of algorithms which utilize Gaussian scale-spaces (Lowe, 2004).

**Neural ODEs.** Studies have shown that both the contrast (Albrecht et al., 2002) and spatial frequency (Frazor et al., 2004) response functions of cortical neurons display characteristic temporal profiles. However, temporal dynamics are not incorporated into typical feed-forward CNN models. In addition, it has been suggested that lateral interactions play an important role in the generation of complex and selective neuronal responses (Angelucci & Bressloff, 2006). Such activity dynamics are often computationally modeled using recurrently coupled neuronal populations whose activations evolve according to coupled differential equations (Ben-Yishai et al., 1995; Ernst et al., 2001).

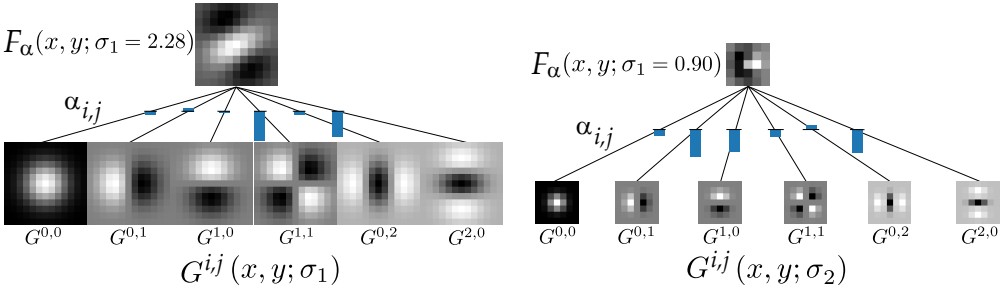

Figure 1: SRF filters based on N-jet filter approximation. Convolutional filters are defined as the weighted sum of Gaussian derivative basis functions up to order 2, with corresponding scales $\sigma_1 = 2.28$ (left) and $\sigma_2 = 0.90$ (right). Our DCN models learn both the coefficients $\alpha$, and the scale $\sigma$ end-to-end during training.

For temporally continuous computational representations consistent with biological models, we adopt the framework of neural ODEs (Chen et al., 2018). Neural ODE interpretation of ResNet models presents an opportunity to explicitly model the temporal dynamics of feature extractors in DNNs. Under certain assumptions, neural ODEs can be interpreted as recurrent interactions (Rousseau et al., 2019), which are biologically plausible (Liao & Poggio, 2016). Unlike ODEs based on pixel-based convolutional filters, the combination with structured filters (SRFs) also provide an intuitive way to parametrize the evolution of the kernels as a function of depth.

Ultimately, we see spatio-temporally continuous representations in end-to-end trainable networks as a link between modern CNN architectures and computational models of biological vision. Although for small spatio-temporal scales it may be more appropriate to use discrete descriptions of biological neurons, such as photoreceptors or populations located in spatially discrete locations, or to consider temporally distinct spiking dynamics, computational models using continuous population activity or rate-based models provide reasonably good explanations of phenomena observed at the network or systems level (Ben-Yishai et al., 1995; Dayan & Abbott, 2001). We believe such larger scale computational models align well with the purposes of computer vision.

## 2.2 STRUCTURED RECEPTIVE FIELDS

We use the multiscale local N-jet formulation (Florack et al., 1996) to define the filters in convolutional layers. Structured receptive fields (SRFs) based on the Gaussian N-jet basis functions are highly applicable to CNNs, as they represent a Taylor expansion of the input image or feature maps in a local neighbourhood in space and scale, and can be used to approximate pixel-based filters (Appendix A.1). This means that each filter $F(x, y; \sigma)$ in the network is a weighted sum of $N$ basis functions, which are partial derivatives of the isotropic two-dimensional Gaussian function $G(x, y; \sigma) = \frac{1}{2\pi\sigma^2} e^{\frac{-(x^2+y^2)}{2\sigma^2}}$. The scale, or the spatial extent, of the filter is explicitly modelled in the $\sigma$ parameter of the Gaussian, which also indirectly determines the spatial frequency response of the SRF (Figure 1).

Formally, the N-jet formulation of an SRF filter $F(x, y)$ is given by:

$$F_\alpha(x, y; \sigma) = \sum_{\substack{0 \le i, 0 \le j}}^{i+j \le N} \alpha_{i,j} \, G^{i,j}(x, y; \sigma) = \sum_{\substack{0 \le i, 0 \le j}}^{i+j \le N} \alpha_{i,j} \frac{\partial^{i+j}}{\partial x^i \partial y^j} G(x, y; \sigma), \quad (1)$$

where $G^{i,j}(x, y; \sigma)$ are the partial derivatives of the Gaussian $G(x, y; \sigma)$ with respect to $x$ and $y$, $N$ is the degree of the Taylor polynomial which determines the basis order, and $\alpha$ encodes the expansion coefficients.

Defined this way, SRFs have favourable properties over pixel-based filters. SRF filters are steerable by the coefficients $\alpha$ and the basis functions are spatially separable. Likewise, due to their spatially continuous description, the filters can be trivially scaled, or rotated, without interpolation.

Figure 1 shows the N-jet approximation of two filters in two different scales $\sigma_1$ and $\sigma_2$. We note that both the coefficients $\alpha$ and the scale $\sigma$ are learnable filter parameters. Instead of fixing the scale $\sigma$ *a priori* and optimizing for $\alpha$ as in Jacobsen et al. (2016) and Sosnovik et al. (2020), we integrate

both these parameters in the network optimization, thus learning not only the shape but also the spatial support of the filters.

## 2.3 NEURAL ODEs

We model the continuous evolution of feature maps within an 'ODE block'. Formally, an ODE block contains a stack of $M$ convolutional layers, each with its own convolutional filters $\mathbf{w}^m$ with $m = 1 \dots M$, followed by normalization $G_{norm}(\cdot)$ and non-linear activation $CELU(\cdot)$ functions. Following the notations of Chen et al. (2018) and Ruthotto & Haber (2019), we define the equations of motion for the feature states $\mathbf{h} \in \mathbb{R}^n$ as:

$$\frac{d\mathbf{h}(t)}{dt} = f(\mathbf{h}(t), t, \mathbf{w}^m, \mathbf{d}^m) = G_{norm} \left[ \mathbf{K}_2(\mathbf{w}^2) g(\mathbf{K}_1(\mathbf{w}^1) g(\mathbf{h}) + \mathbf{J}_1(\mathbf{d}^1) t) + \mathbf{J}_2(\mathbf{d}^2) t \right] \quad (2)$$

where $g(\mathbf{x}) = CELU(G_{norm}(\mathbf{x}))$, the linear operators $\mathbf{K}_m \in \mathbb{R}^{n \times n}$ denote the convolution operators parametrized by $\mathbf{w}^m$, and $\mathbf{J}_m \in \mathbb{R}^n \times \mathbb{R}$ denote the linear explicit $t$-terms parametrized by $\mathbf{d}^m$. The filters $\mathbf{w}^m, \mathbf{d}^m(\theta)$ are functions of some learnable parameters $\theta$. As a typical CNN convolution operator, $\mathbf{K}_m(\mathbf{w}^m)$ with 2 input and 2 output channels can be written as

$$\mathbf{K}_m(\mathbf{w}^m) = \begin{pmatrix} \mathbf{K}_m^{1,1}(\mathbf{w}_{1,1}^m) & \mathbf{K}_m^{1,2}(\mathbf{w}_{1,2}^m) \\ \mathbf{K}_m^{2,1}(\mathbf{w}_{2,1}^m) & \mathbf{K}_m^{2,2}(\mathbf{w}_{2,2}^m) \end{pmatrix} \quad (3)$$

with $\mathbf{w}_{ji}^m$ the convolutional kernels for input channel $i$ and output channel $j$ of the $m$-th convolution. In this definition, the discrete depth of feed-forward networks such as ResNets is reimagined as a continuous dimension defined by time $t$, where the input image defines the initial conditions $\mathbf{h}(0)$. For the rest of this paper, we stick to the original ODE-Net interpretation that the number of function evaluations performed by the numerical ODE solver is analogous to network depth.

In accordance with conventional ResNet blocks, we pick $M = 2$. Based on the original implementation by Chen et al. (2018), $G_{norm}$ is defined as group normalization (Wu & He, 2018). For generalized compatibility with neural ODEs and the adjoint method, we choose a non-linear activation function with a theoretically unique and bounded adjoint, namely continuously differentiable exponential linear units, or CELU (Barron, 2017). Similarly, we keep the linear dependence of the equations of motion on time (or network depth) $t$. Finally, we adapt the GPU implementation of ODE solvers[1] to solve the equations of motion for a predefined time interval $t \in [0, T]$ using the adaptive step size DOPRI method.

## 2.4 DEEP CONTINUOUS NETWORKS WITH SRFs AND NEURAL ODEs

We formulate deep continuous networks (DCNs) by employing learnable, continuous SRF filter descriptions to define the weights in the evolution of a neural ODE. This means that for DCNs, each $\mathbf{w}_{ji}^m$ in Eq. 2 is a discretization of the continuous SRF filter $F_\alpha(x, y; \sigma)$ given in Eq. 1, sampled in $[-2\sigma, 2\sigma]$. We define the spatial axes such that the width and height of a pixel is $\delta_x = \delta_y = 1$ and the sampling rate is 1 ($\delta_x^{-1}$). $\alpha_{ji}^m$ and $\sigma^m$ are trainable filter parameters, where $\sigma_{ji}^m = \sigma^m$ is shared between the filters in a convolutional layer unless stated otherwise. All our code is available at[2].

**Network architecture and training.** We construct DCNs by stacking ODE blocks separated by downsampling blocks (Figure 2). Each downsampling block is a sequence of normalization, non-linear activation and strided convolution. We use a convolutional layer for increasing the channel dimensionality at the input level and employ global average pooling and a fully connected layer at the output level. We train our networks using cross entropy loss and the CIFAR-10 dataset (Krizhevsky, 2009). (See Appendices A.2-A.3 for further details regarding training parameters.)

As a baseline without spatial continuity, we compare DCN performance to the ODE-Net introduced in Chen et al. (2018), where the convolutions within the ODE blocks are performed using discrete, pixel-based kernels, with $k \times k$ parameters. As a baseline without temporal continuity, we define the 'ResNet-blocks' model where the ODE blocks are replaced by generic, discrete ResNet blocks, comprising two convolutional layers and a skip connection, with comparable number of parameters

---

[1]https://github.com/rtqichen/torchdiffeq/
[2]code will be made available at github

to the ODE-Net. This is similar to the baseline model used in Chen et al. (2018). In the ResNet-SRF-blocks model, we provide the discrete-time and continuous-space baseline by replacing the $k \times k$ filter definition of ResNet-blocks with SRF definitions.

We test two versions of DCNs and ResNet-SRF-blocks to quantify the viability of SRF filters outside of the ODE blocks. In DCN-ODE and ResNet-SRF-blocks we use the SRF filters only within the ODE (ResNet) blocks, and for the remaining layers we use discrete kernels with the same hyperparameters as the baselines. In the second version, DCN-full (ResNet-SRF-full), we use spatially continuous kernels everywhere, including the downsampling layers.

Finally, we investigate the case where we drop scale sharing within a layer, and optimize the scale parameter $\sigma_{ji}^m$ independently for each input channel $i$ and output channel $j$, which we call DCN $\sigma_{ji}$.

**Meta-parametrization.** DCNs enable us to parametrize the trainable filter parameters $\alpha$ and $\sigma$ as a function of time $t$. This both enables the kernels to vary smoothly over time, and lets us define temporal dynamics for the neuronal responses in our network. We test the viability of such models by introducing DCN variants where $\sigma$ and/or $\alpha$ are defined using linear or quadratic functions of $t$ and learnable parameters $a$, $b$, $c$, $a_s$, $b_s$, $a_\alpha$ and $b_\alpha$ (Table 3).

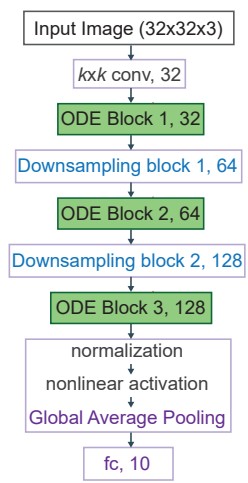

Figure 2: DCN model architecture with CIFAR-10 input images. Downsampling is performed via convolutions with stride 2. The convolutional kernel size $k$ depends on the model, and is learned during training in DCNs. The equations of motion (Eq. 2) are solved within ODE blocks.

## 3 EXPERIMENTAL ANALYSIS

### 3.1 PARAMETER REDUCTION AND DATA EFFICIENCY

Similar to biological models, where analytical receptive fields limit the scope of the model using prior information, we find that DCNs are more parameter efficient compared to baseline networks. Evaluated on CIFAR-10, DCNs perform on par with baselines, despite using SRFs of a small basis order 2, which means each filter shape is defined by only 6 free parameters as opposed to 9 for the conventional $3 \times 3$ kernels (Table 1). In addition, we find that parameter reduction via the use of SRFs with a small basis order also leads to data efficiency. When trained on a subset of CIFAR-10 images (small-data regime), DCNs outperform the discrete baseline networks (Table 2a).

As an additional demonstration of the versatility of the spatio-temporal representations learned by DCNs, we perform an image reconstruction experiment. We use the feature maps generated by the networks trained on the CIFAR-10 classification task (output of ODE Block 3 in Fig. 2), as input to a decoder network. The decoder networks are composed of 2 DCN-ODE, ODE-Net or ResNet blocks, separated by bilinear upscaling layers and $1 \times 1$ convolutions to reduce the output dimensionality

| Model | Continuity | | Accuracy (%) | # Parameters |
| | Spatial | Temporal | | |
|---|---|---|---|---|
| ResNet-blocks | x | x | $89.01 \pm 0.17$ | 555K |
| ResNet-SRF-blocks | ✓ | x | $88.26 \pm 0.03$ | 426K |
| ResNet-SRF-full | ✓ | x | $89.32 \pm 0.39$ | 323K |
| ODE-Net (Chen et al., 2018) | x | ✓ | $89.60 \pm 0.28$ | 560K |
| DCN-ODE | ✓ | ✓ | $89.46 \pm 0.16$ | 429K |
| DCN-full | ✓ | ✓ | $89.18 \pm 0.32$ | 326K |
| DCN $\sigma_{ji}$ | ✓ | ✓ | $89.74 \pm 0.30$ | 472K |

Table 1: CIFAR-10 validation accuracies of DCN models, averaged over 3 runs, compared to baseline models. ODE-Net and ResNet-blocks baselines are as introduced in Chen et al. (2018). DCNs perform on par with spatially and/or temporally discrete baselines, despite having a lower number of trainable parameters.

| Model | # images per class | | | | Model | Reconstruction Loss (%) |
|---|---|---|---|---|---|---|
| | 52 | 103 | 512 | 1024 | | |
| ResNet-blocks | $39.8 \pm 0.6$ | $49.0 \pm 0.2$ | $70.4 \pm 1.2$ | $76.8 \pm 0.7$ | ResNet-blocks | $21.0 \pm 0.4$ |
| ODE-Net | $41.7 \pm 1.2$ | $48.6 \pm 0.5$ | $71.7 \pm 1.5$ | $77.4 \pm 0.5$ | ODE-Net | $20.2 \pm 1.3$ |
| DCN-ODE | $\mathbf{44.5} \pm 0.8$ | $\mathbf{54.2} \pm 0.8$ | $\mathbf{75.5} \pm 0.8$ | $\mathbf{79.7} \pm 0.3$ | DCN-ODE | $\mathbf{17.1} \pm 0.3$ |
| | (a) | | | | | (b) |

Table 2: (a) Validation accuracies on a subset of CIFAR-10 (small-data regime) for the DCN-ODE model versus baselines when the training data is limited. The DCN model outperforms spatially and/ or temporally discrete baselines as parameter efficiency leads to data efficiency. (b) DCNs achieve lower MSE loss in the reconstruction task than baselines on the CIFAR-10 validation set, despite using a smaller number of parameters. All results are averaged over 3 runs.

to three (RGB). We train the networks to reconstruct CIFAR-10 images using mean squared error (MSE) loss. We find that the DCN models outperform discrete baseline models on the validation set (Table 2b), despite having a lower number of parameters as before. Additional details and example images are shown in Appendix A.4.

Similarly, we find that meta-parametrized DCN variants match the classification performance of baselines and may outperform DCNs with static weights (Table 4). This is an interesting finding since we test only a few models, with little hyperparameter optimization, which indicates that DCNs can potentially be used to parametrize the dependence of convolutional kernel weights on network depth, for further parameter reduction.

## 3.2 LINK WITH BIOLOGICAL MODELS

**Scale fitting.** As an advantage over conventional CNNs, it is possible to directly investigate the optimal receptive field (RF) size in each DCN block after training, since DCNs fit the kernel scale $\sigma$ explicitly. We observe an upward trend in the SRF scale $\sigma$ with the depth of the convolutional layer within the network (Fig. 3a). While the RF size grows with depth also in conventional CNNs, it typically grows linearly over convolutions, as the kernel size is constant across layers. This is a limitation in most CNNs that the visual system does not necessarily have. DCNs, on the other hand, can learn RF sizes which grow non-linearly as a function of depth, which seems to be in line with the behaviour in downstream visual areas (Smith et al., 2001). In addition, we plot the distribution of learned $\sigma_{ji}$ in different ODE blocks of the model DCN $\sigma_{ji}$ (Fig. 3b). Note that the scale parameter $\sigma$ controls the bandwith of the SRF filters and is thus related to their spatial frequency response. We find that the $\sigma_{ji}$ distributions after training are approximately log-normal and display a positive skew, which is consistent with the scale and spatial frequency tuning distributions in the primate visual system (Yu et al., 2010). We believe these results are promising for bridging the gap between deep learning and traditional models of biological systems.

**Pattern completion.** Established models from computational neuroscience, with continuous temporal dynamics and well-defined recurrent interaction structures, such as the Ermentrout-Cowan model (Bressloff et al., 2001), or neural field models (Amari, 1977), display interesting high-level phenomena such as spontaneous pattern formation and travelling waves (Coombes, 2005). Such models employ local, distance-dependent interactions, similar to the SRF based ODE blocks in the DCN formulation. Based on this resemblance, we hypothesize that DCNs can perform well in the case of locally missing information in images, through pattern completion at the feature map level.

| Model | Parametrization |
|---|---|
| DCN $\sigma(t)$ | $\sigma = 2^{at+b}$ |
| DCN $\sigma(t^2)$ | $\sigma = 2^{at^2+bt+c}$ |
| DCN $\sigma(t), \alpha(t)$ | $\sigma = 2^{a_s t + b_s}, \alpha = a_\alpha t + b_\alpha$ |

Table 3: Meta-parametrization of filter parameters $\sigma$ and $\alpha$ as a function of time $t$ in different DCN variants.

| Model | Accuracy (%) |
|---|---|
| DCN $\sigma(t)$ | $89.97 \pm 0.30$ |
| DCN $\sigma(t^2)$ | $89.93 \pm 0.28$ |
| DCN $\sigma(t)$ and $\alpha(t)$ | $89.88 \pm 0.25$ |

Table 4: CIFAR-10 validation accuracies for DCN models with meta-parametrization.

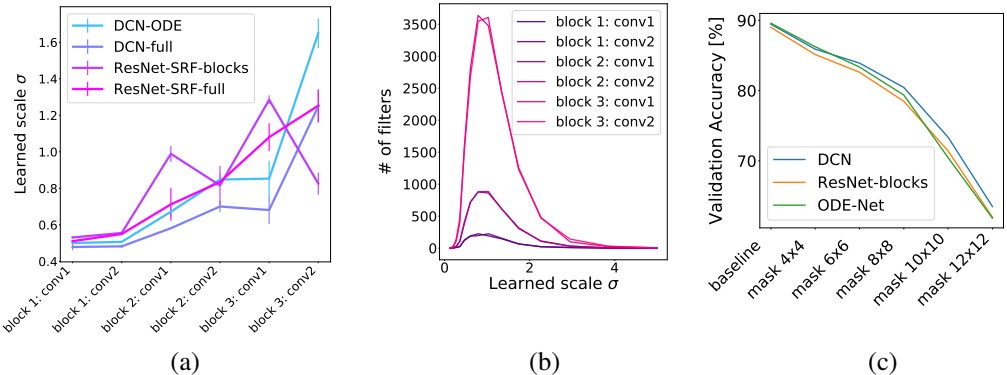

Figure 3: (a) Learned $\sigma$ values increase with depth within the network. (b) $\sigma_{ji}$ distributions within the ODE blocks display a positive skew in line with biological observations. (c) CIFAR-10 validation accuracies on the pattern completion task with increasing mask size.

We test this hypothesis by masking $n \times n$ pixels on the CIFAR-10 validation set at test time. The masks have zero pixel values, and are placed at the center of the image. We find that when confronted with a small patch of missing information, DCNs can generate feature maps similar to those obtained from intact images. Specifically, we observe that the difference $D(t) = \frac{1}{A} \sum |\mathbf{h}^{\text{im}}(t) - \mathbf{h}^{\text{im\_masked}}(t)|$ between the feature maps generated by an intact image $\mathbf{h}^{\text{im}}(t)$ and a masked image $\mathbf{h}^{\text{im\_masked}}(t)$, normalized by the amplitude of the intact image $A$, is reduced within an ODE block (Figure 4a). In terms of the overall classification performance with masked images at test time, we find that DCNs are marginally more robust against zero-masking than baselines (Fig. 3c).

### 3.3 CONTRAST ROBUSTNESS AND COMPUTATIONAL EFFICIENCY

The selectivity of neuronal responses is invariant to contrast in mammalian vision (Sclar & Freeman, 1982; Skottun et al., 1987). However, we observe that DCN and ODE-Net models are sensitive to changes in input contrast. This is not unexpected since ODE blocks compute the solution to the initial value problem posed by the equations of motion and the input feature maps, and contrast variations change the initial conditions. To quantify this sensitivity we vary the contrast $c$ of the input images at test time, where for each image $H_i$ in the CIFAR-10 validation set we define the input as $H_j = cH_i$. When naively changing the input contrast $c$ this way, we find that the validation accuracy decays rapidly for both models (solid lines in Fig. 4b, top).

Empirically, we notice that, with the appropriate choice of normalization functions, the input contrast $c$ has a direct effect on the time scales of the solution $\mathbf{h}(t)$. Based on this observation, we heuristically test whether scaling the integration time interval $T$ (used during training) of ODE block 1 by $c$ for each input image $j$, $T_j = cT$, can improve contrast robustness at test time. We find that with the scaled integration interval DCN validation accuracy is relatively robust against changes in contrast $c$, compared to naive baselines and ODE-Net, until $c << 1$ when time scales become too fast and the ODE solver becomes unstable for all models (dashed lines in Fig. 4b, top).

Interestingly, we observe a reduction in the number of function evaluations (NFEs) in ODE block 1 for $c < 1$ (Fig. 4b, middle). Furthermore, we show that as long as the error tolerance of the ODE blocks are not decreased, this effect can be exploited by scaling the input feature maps of all ODE blocks by $c$ for significant computational savings. We find that decreasing $c$ leads to considerable efficiency improvements, where total NFEs can be reduced from 102 to 60, (for $c = 1$ and 0.06), with less than 0.5% loss in accuracy (Fig. 4b, bottom).

## 4 RELATED WORK

**Spatially continuous filter representations.** Structured filters have been traditionally used in computer vision for extracting image structure at multiple scales. N-jet filter basis is first introduced by Florack et al. (1996) based on previous work on Gaussian scale-spaces (Florack et al., 1992; Lindeberg, 2013). We use the N-jet basis in order to approximate convolutional filters, as they enable a spatially continuous representation with an explicit scale parameter $\sigma$.

Similar to the N-jet basis, a set of oriented multi-scale wavelets, called a steerable pyramid, is proposed by Simoncelli et al. (1992) and complex wavelets have been used by Bruna & Mallat

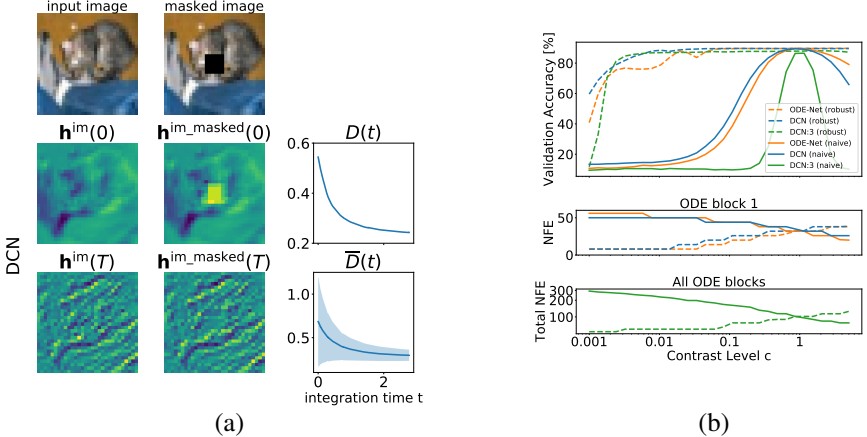

Figure 4: (a) Pattern completion in the DCN feature maps for an example image. Feature maps are shown for a single channel in ODE block 1. We find that the difference $D(t)$ between the feature maps $\mathbf{h}^{\text{im}}(t)$ of an intact image and $\mathbf{h}^{\text{im\_masked}}(t)$ of a masked image is reduced as $t \to T$. We also show the mean $\overline{D}(t)$ for 1000 validation images (bottom right), where the shaded area is the standard deviation over different images. Example feature maps from other models are provided in Appendix A.5. (b) Top: In terms of CIFAR-10 validation performance, DCNs are more robust than baseline ODE-Nets to changes in input contrast $c$ at test time. Interestingly, the number of function evaluations (NFEs) in the first ODE block (middle) or the whole DCN network (bottom) can be reduced considerably by modulating $c$.

(2013) and Mallat (2012) as part of scattering transforms. CNN filters based on linear combinations of Gabor wavelets are adopted by Luan et al. (2017), while Worrall et al. (2017) propose circular harmonics, as spatially continuous filter representations.

Similar to our approach, Shelhamer et al. (2019) combine free-form filters with Gaussian kernels, thus learning the filter resolution. Likewise, Xiong et al. (2020) learn filter sizes using Gaussian kernels optimized using variational inference. Here, we use the N-jet framework based on Gaussian derivatives as in Jacobsen et al. (2016), however our main motivation is retaining compatibility with biological models. Also, unlike Jacobsen et al. (2016) we learn the scale parameter $\sigma$ during training.

**Continuous time representations in deep networks.** Along with work by Lu et al. (2018) and Ruthotto & Haber (2019), networks continuous in the temporal (or depth) dimension have been proposed by Chen et al. (2018) under the name neural ordinary differential equations (ODEs). They propose ODE-Nets based on the ResNet formulation (He et al., 2016) for classification tasks, which is used as a baseline in this paper. In this work we focus mainly on an image classification task, however, there is extensive ongoing work on generative models and normalizing flows using the neural ODE continuous time interpretation (Salman et al., 2018; Grathwohl et al., 2019). We note that DCNs can be readily incorporated into continuous flow models, as well as other spatio-temporally continuous DNN interpretations based on partial differential equations (Ruthotto & Haber, 2019).

Even though the adjoint method described in Chen et al. (2018) offers considerable computational savings, especially in terms of memory, recent work has improved upon it both in terms of stability, computational efficiency and performance (Dupont et al., 2019; Finlay et al., 2020; Zhuang et al., 2020b). Likewise, the contrast robust formulation of DCNs, as well as the synergy between the $\mathcal{O}(1)$ memory complexity of the adjoint method and spatially separable SRF filters (the implementations of which may otherwise inflate the memory cost) provide potential computational benefits over conventional CNNs where the number of function evaluations is fixed.

Other studies have suggested that, similar to our DCN variants where the filter definitions are independent of time, neural ODEs based on ResNet architectures with weight sharing can be interpreted as recurrent neural networks (Rousseau et al., 2019; Kim et al., 2016), which bridges the gap between deep learning, dynamical systems and the primate visual cortex (Massaroli et al., 2020; Liao & Poggio, 2016). Similar to these works, we capitalize on the parallels between neural ODEs and the dynamical systems approach of the computational models of recurrent, biological circuits in order to extend them to fully continuous networks, where not only the depth of the network is continuous but also the shape and spatial resolution of the filters are end-to-end trainable.

**CNNs and RNNs as models of biological networks.** There is extensive prior work on CNNs and recurrent neural networks (RNNs) for modeling biological computation. The visual cortex is highly recurrent (Dayan & Abbott, 2001; Liao & Poggio, 2016) which is thought to be responsible for complex neuronal dynamics (Ben-Yishai et al., 1995; Angelucci & Bressloff, 2006). Accordingly, computational models with lateral connections (Sompolinsky et al., 1988; Ernst et al., 2001) and more recently RNNs (Laje & Buonomano, 2013; Mante et al., 2013; Mastrogiuseppe & Ostojic, 2018) have been extensively used as models of biological neural computation. For example the first-order reduced and controlled error (FORCE) algorithm, have been used to reproduce the dynamics of different biological circuits (Sussillo & Abbott, 2009; Laje & Buonomano, 2013; Carnevale et al., 2015; Rajan et al., 2016; Enel et al., 2016). Similarly, optimization via gradient-based algorithms such as the Hessian-free method (HF) or stochastic gradient-descent (SGD) have been adopted to train recurrent networks (Mante et al., 2013; Barak et al., 2013; Song et al., 2016) to replicate experimental observations. It has also been suggested to use spiking recurrent networks (Kim & Chow, 2018; Kim et al., 2019) and incorporate synaptic dynamics (Ba et al., 2016; Miconi et al., 2018) for improved physiological realism. Finally, recurrent convolutional networks (RCNNs) have been proposed (Liang & Hu, 2015; Spoerer et al., 2017; Hu & Mihalas, 2018), which can emulate biological lateral connectivity structures and extra-classical receptive field effects.

In contrast to typical RNNs, our model is based on the ResNet inspired model of neural ODEs, and in its current form (Eq. 2), does not accept time-variant input. In that sense, the spatio-temporal dynamics of DCNs refer to the dynamics of the neurons and not the input. Although with weight sharing DCNs can be thought of as recurrent networks (Rousseau et al., 2019) and can be modified to process time-variant input (such as videos), in this paper we consider the DCN models to be an extension of conventional feed-forward CNNs, with extended temporal dynamics and continuous spatial representations, which are applicable to feed-forward models of the visual system similar to works by Lindsey et al. (2019); Ecker et al. (2019); Schrimpf et al. (2018); Zhuang et al. (2020a).

## 5 DISCUSSION

We introduce DCNs, CNN models which learn spatio-temporally continuous representations, consistent with biological models. We showed that DCNs can match baseline performance in an image classification task and outperform baselines in the small-data regime and in a reconstruction task, while using a smaller number of parameters. Similarly, we have proposed different methods of meta-parametrization of the convolutional filter as a function of time, which may not only be viable for network compression purposes, but also for modelling the temporal profiles of biological responses. As a further link with biological models, we have demonstrated that the learned scale distributions in DCNs are compatible with experimental observations, which seems promising for the applicability of DCN models to future neuroscientific investigations regarding the emergence of RF sizes. In addition, we have presented the capability of DCNs to reduce errors in feature maps caused by masking. Finally, we have empirically shown an interplay between the input contrast to ODE blocks and the time scales of the solutions, which can be capitalized on for computational savings.

However, one of the biggest limitations of DCNs is that they may become unstable during training. This is a combined problem of neural ODE models and scale fitting, which may lead to exploding filter sizes with larger learning rates. Especially for meta-parametrization, it would be advisable to clip the integration time and filter parameters within a reasonable range. Nevertheless, we believe there are exciting future research opportunities involving DCNs. A rigorous mathematical analysis of the nonlinear system of ODEs is beyond the scope of this paper, however, neural ODE formulations provide interesting opportunities for establishing a theoretical understanding of DNNs based on dynamical systems. The interplay of input contrast and integration time is one such observation which requires further scrutiny. Similarly, our choice of filters based on well-behaved Gaussian derivatives allow for further analytical investigations, unlike conventional CNNs.

Similarly, DCNs offer interesting possibilities for biological modelling. Although there is still a large gap between modern CNNs, DCNs and the complex biological realism, the inbuilt smooth evolution of filters in DCNs can be used, for example, to incorporate response dynamics such as synaptic depression or short-term potentiation (similar to previous work by Ba et al. (2016); Miconi et al. (2018)). Likewise, the equations of motion can be modified to reflect axonal delays or generate oscillations. In short, we believe by offering a link between dynamical systems, biological models and CNNs, DCNs display an interesting potential to bring together ideas from both fields.

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

## A    APPENDIX

### A.1    GAUSSIAN MULTISCALE LOCAL N-JET

Based on the Schwartz theory of smooth test functions, the Gaussian scale-space paradigm states that the derivatives $L_{i_1 \dots i_n}(\vec{x}; \sigma)$ of a function $L_0(\vec{x})$ with respect to the spatial variables $x_i$, with $i = 1 \dots d$, and at scale $\sigma$ is given by the convolution

$$L_{i_1 \dots i_n}(\vec{x}; \sigma) = L_0 * \partial_{i_1 \dots i_n} G(\vec{x}; \sigma) \qquad (4)$$

where $\partial_{i_1 \dots i_n}$ is the $n$-th order partial derivative and $G(\vec{x}; \sigma)$ is the normalized, isotropic Gaussian kernel with standard deviation $\sigma_i = \sigma$ and mean $\mu_i = 0$ (Florack et al., 1996). Note that $L_0(\vec{x})$ does not need to be a smooth function, and therefore the Gaussian scale-space paradigm can be applied to obtain local image derivatives in different scales, where $\vec{x}$ denote the spatial coordinates and $\sigma$ can be interpreted as the coordinate in the scale dimension.

We build upon this definition of local image derivatives to build our structured receptive fields (SRFs), similar to Jacobsen et al. (2016). The main idea we leverage is that using a Taylor approximation, one can decompose an input image into a superimposition of its local derivatives. This decomposition can then be performed by local convolution kernels in CNNs, where the relative weight of each derivative order can be learned during training. In order to show this, we observe that the $N$-th order Taylor expansion of an image $L : \mathbb{R}^2 \to \mathbb{R}$ around a point $(a, b)$ is given by

$$L(x, y) = \sum_{i=0}^{N} \sum_{j=0}^{N-i} \frac{(x-a)^i (y-b)^j}{i! j!} \frac{\partial^{i+j}}{\partial x^i \partial y^j} L(a, b) \qquad (5)$$

where the partial derivatives of $L$ with respect to $x$ and $y$ can be interpreted as $L_{1_1 \dots 1_n}(x, y; \sigma_0)$ and $L_{2_1 \dots 2_n}(x, y; \sigma_0)$ from equation 4 at some original scale $\sigma_0$. This means that, via the linearity of convolution, and under the assumption that $L(x, y; \sigma)$ does not diverge, it is equivalent to either use the $N$-th order derivatives of the input image, or use the $N$-th order derivatives of the Gaussian function $G(\vec{x}; \sigma)$, to perform the decomposition at scale $\sigma$. The local N-jets can thus be seen to be parametrized by the coefficients in the expansion in equation 5. By definition, we consider the Taylor expansion coefficients to be covariant derivatives of the image $L(x, y; \sigma)$, which are independent

of the local coordinate system, to reach the filter approximations given in equation 1, where the coefficients take the form $\alpha_{i,j}$.

In short, the multiscale local N-jet provides a natural decomposition of an image in a local neighborhood in the spatial and scale dimensions. Under convolution with $G(\vec{x}; \sigma)$, this decomposition provides a framework for defining convolutional filters in a CNN using $N$-th order Taylor polynomials. The SRF filters we use are based on this N-jet definition and allow us to learn the scale, spatial frequency and orientation of the filters during training, which are fundamental properties of biological receptive fields (Jones & Palmer, 1987; Lindeberg, 1993).

In addition, the Gaussian scale-space formulation of SRFs (Jacobsen et al., 2016) lead to theoretically interesting properties which strengthen the motivation for our choice of filters based on the Gaussian N-jet. For example, the semi-group property of Gaussian scale-spaces indicates for a Gaussian derivative kernel $G^{i,j}(x, y; t)$ parametrized by the variance $t = \sigma^2$

$$G^{i,j}(x, y; t + t') = G^{i,j}(x, y; t) * G(x, y; t') \tag{6}$$

where the superscripts $(i, j)$ denote the partial derivatives with respect to $x$ and $y$. This means that a translation in scale dimension can be achieved through convolution with a simple (0-th order) Gaussian kernel. For DCNs, this means that we have a solid understanding of the scale of the feature maps (in the sense of the Gaussian scale-space) at every layer in the network, as long as we know the learned value of the $\sigma$s. In the absence of SRFs with an explicit scale parameter, this information is lost.

Similarly, SRF filters based on Gaussian derivatives are steerable by the coefficients $a_{i,j}$. For example, a filter with orientation $\theta$ can be described using second order basis functions as

$$G_\theta^{2,0} = a_{2,0}G^{2,0} + a_{1,1}G^{1,1} + a_{0,2}G^{0,2} = \cos^2(\theta)G^{2,0} - 2\cos(\theta)\sin(\theta)G^{1,1} + \sin^2(\theta)G^{0,2}. \tag{7}$$

Finally, the set of Gaussian derivative basis functions $G^{i,j}(x, y; \sigma)$ are spatially separable

$$G^{i,j}(x, y; \sigma) = G^i(x; \sigma)G^j(y; \sigma) \tag{8}$$

which is useful for computational efficiency in numerical applications.

## A.2 TRAINING PROCEDURE

The basic architecture of all our models is given in figure 2. Unless otherwise stated, in all models we use group normalization (Wu & He, 2018) with 32 groups in every layer as the normalization function. As the nonlinear activation, we use continuously differentiable exponential linear units (Barron, 2017), or CELU for DCN models. Based on the original ODE-Net implementation (Chen et al., 2018), for ODE-Net models and ResNet-block models, we use rectified linear units (ReLU). Likewise, when defining the integration time interval $T$, we stick to the original implementation, with $T = 1$ for ODE-Net models, whereas we use $T = 2$ for DCN models. Otherwise, all the hyperparameters are kept constant between models. For a brief overview of hyperparameter optimization, see appendix A.3.

All models are trained for 100 epochs on the standard CIFAR-10 training set (Krizhevsky, 2009) using cross-entropy loss. As data augmentation, we use random translations up to 4 pixels in each spatial dimension and random horizontal flips. Optimization is performed using SGD with a mini-batch size of 128, initial learning rate $10^{-1}$, momentum 0.9, and a learning rate decay by a factor of 0.1 at epochs 40 and 70. For continuous time models based on neural ODEs, we use the adjoint method for backpropagating the losses and ODE solvers with error tolerance set to $10^{-3}$.

For convolutional layers with pixel-based $k \times k$ filters, the weights are initialized using the standard Kaiming uniform initialization (He et al., 2015). For layers using SRF filters, the initial $\alpha$ values are randomly sampled from a normal distribution with mean 0 and standard deviation 0.1, and initial $\sigma$ values are sampled from a normal distribution with mean 0 and standard deviation 2/3.

For the restricted CIFAR-10 experiments (small-data regime), we pick the total number of training images to match our mini-batch size of 128, or otherwise have a minimal number of samples in the final batch (which is dropped in each epoch). In order to confirm convergence, for the training set sizes $[520, 1030, 5120]$, we increased the number of training epochs by a factor of $[10, 5, 2]$ respectively.

For meta-parametrized models the initial values for the learnable parameters follow normal distributions $\mathcal{N}(\mu, \sigma_{\mathcal{N}})$: $a \sim \mathcal{N}(0, 2/3)$, $b \sim \mathcal{N}(0, 0.1)$, for the DCN $\sigma(t)$ model; $a \sim \mathcal{N}(0, 2/3)$, $b \sim \mathcal{N}(0, 2/3)$, $c \sim \mathcal{N}(0, 0.1)$ for the DCN $\sigma(t^2)$ model; and $a_s \sim \mathcal{N}(0, 2/3)$, $b_s \sim \mathcal{N}(0, 0.1)$, $a_\alpha \sim \mathcal{N}(0, 0.1)$ and $b_\alpha \sim \mathcal{N}(0, 0.05)$ for the DCN $\sigma(t)$, $\alpha(t)$ model.

For all our models using SRF filters, except for the DCN $\sigma_{ij}$ model, we use scale sharing within a convolutional layer such that $\sigma_{ij}^m = \sigma^m$ for all $m$. This makes the GPU implementation trivial, as all filters within a layer are sampled in the interval $[-2\sigma^m, 2\sigma^m]$, and hence the convolutional kernel sizes within a layer are uniform. However, for the DCN $\sigma_{ij}$ model, a GPU implementation would be highly inefficient if we truncated the kernels at a fixed factor of $\sigma_{ij}$, independently for each input and output channel $i, j$. Therefore for the DCN $\sigma_{ij}$ model we fix the kernel size at $7 \times 7$, but we still learn the shape and the scale (bandwidth) of the filters during training.

We use the Dormand–Prince (DOPRI) method (Dormand & Prince, 1980) as the numerical ODE solver. The DOPRI method is an explicit, adaptive solver of the Runge-Kutta family, which uses 6 function evaluations to compute fourth- and fifth-order accurate solutions to ODEs, along with an error estimate. The size of the adaptive steps taken by the solver can be regulated by specifying an error tolerance on this error estimate.

As is the case with most modern ODE solvers, the GPU implementation of the DOPRI solver that we use (Chen et al., 2018) considers the input arguments for the integration time interval $t \in [0, T]$ (or $t \in [0, 2]$ in the case of all DCN models) as soft bounds. This means that the DOPRI algorithm may explore time points outside of this interval, based on its internal error estimation, and may then employ interpolation to return solutions within the specified bounds. In the meta-parametrized models, where for example $\sigma$ is an explicit function of $t$, this may lead to very large or very small kernel sizes, ordinarily unexpected within the integration time interval. In order to avoid numerical instability and memory issues in the meta-parametrized models, we scale down and clip the integration time $t$ when passing it into the parameter definitions as $\sigma(\tau t_{\mathrm{clip}})$ and $\alpha(\tau t_{\mathrm{clip}})$. We clip the $t$ values in the interval $[-0.5, 2.5]$ and use $\tau = 0.5$.

### A.3  HYPERPARAMETER TUNING

As mentioned in appendix A.2, we share all the design choices and hyperparameters between all DCN and baseline ODE-Net models, except for the nonlinear activation function and integration time interval $T$.

This difference in design choices arises since for the ODE-Net baseline we stay faithful to the original ODE-Net implementation, where ReLU is the nonlinear activation function and $T = 1$, whereas for DCN models we use CELU as the activation function, due to its generalized compatibility with the adjoint method, and $T = 2$. In order to verify that our design choices do not provide an unfair advantage over the ODE-Net baseline, we run some control experiments, where we vary the activation function and $T$ in the ODE-Net baseline.

We find that the change of activation functions or integration interval $T$ do not provide a significant increase to the CIFAR-10 classification performance in the ODE-Net baseline (Table 5).

| Model | Accuracy (%) |
|---|---|
| DCN-ODE | $89.46 \pm 0.16$ |
| ODE-Net (baseline) | $89.60 \pm 0.28$ |
| ODE-Net with $T = 2$ | $89.50 \pm 0.07$ |
| ODE-Net with CELU | $89.33 \pm 0.16$ |
| ODE-Net with CELU and $T = 2$ | $89.25 \pm 0.30$ |

Table 5: CIFAR-10 validation accuracy (averaged over 3 runs) in the control experiments testing the effect of DCN model design choices on the baseline ODE-Net.

### A.4 CIFAR-10 IMAGE RECONSTRUCTION

For the reconstruction task, we use the network architectures with 3 DCN-ODE blocks (Fig. 2), or for the baseline networks ODE-Net or ResNet-blocks, trained on the CIFAR-10 classification task as an encoder. We use the feature maps at the end of ODE Block 3 as the input to a small decoder network. The decoder DCN network is made up of an ODE block with 128 input channels, followed by bilinear upscaling, a $1 \times 1$ convolution to reduce the number of channels to 64, another ODE block, followed by bilinear upscaling, normalization, non-linear activation and a $1 \times 1$ convolution to generate the output in RGB space.

We implement reconstruction as a regression task and use the mean squred error (MSE) as the loss function. Otherwise, the training parameters are the same as the classification experiments: We use the SGD optimizer with a mini-batch size of 128, learning rate $10^{-1}$ and momentum 0.9, together with the adjoint method and an error tolerance of $10^{-3}$.

The reconstruction of some example images (randomly selected) from the CIFAR-10 validation set by the DCN and baseline networks are shown in figure 5.

### A.5 PATTERN COMPLETION IN FEATURE MAPS

In figure 6, we show the feature map evolution within the first ODE block (or ResNet block) of different models with and without masking of some example input images. Size of the mask depicted here is $6 \times 6$ pixels and the example images were chosen so as to have the mask located close to the middle of the object. We picked some channels with visible mask-related artifacts in the input feature maps to the first ODE (ResNet) block. Since there is no feature map trajectory in the ResNet-blocks model, but only one input and one output feature map, the difference between the feature maps of the intact and masked image is given as a scatter plot of two data points connected by a red line.

Figure 7 depicts the average difference of intact and masked feature maps $\overline{D}(t)$ averaged over 1000 images and the standard deviation for the DCN and baseline networks.

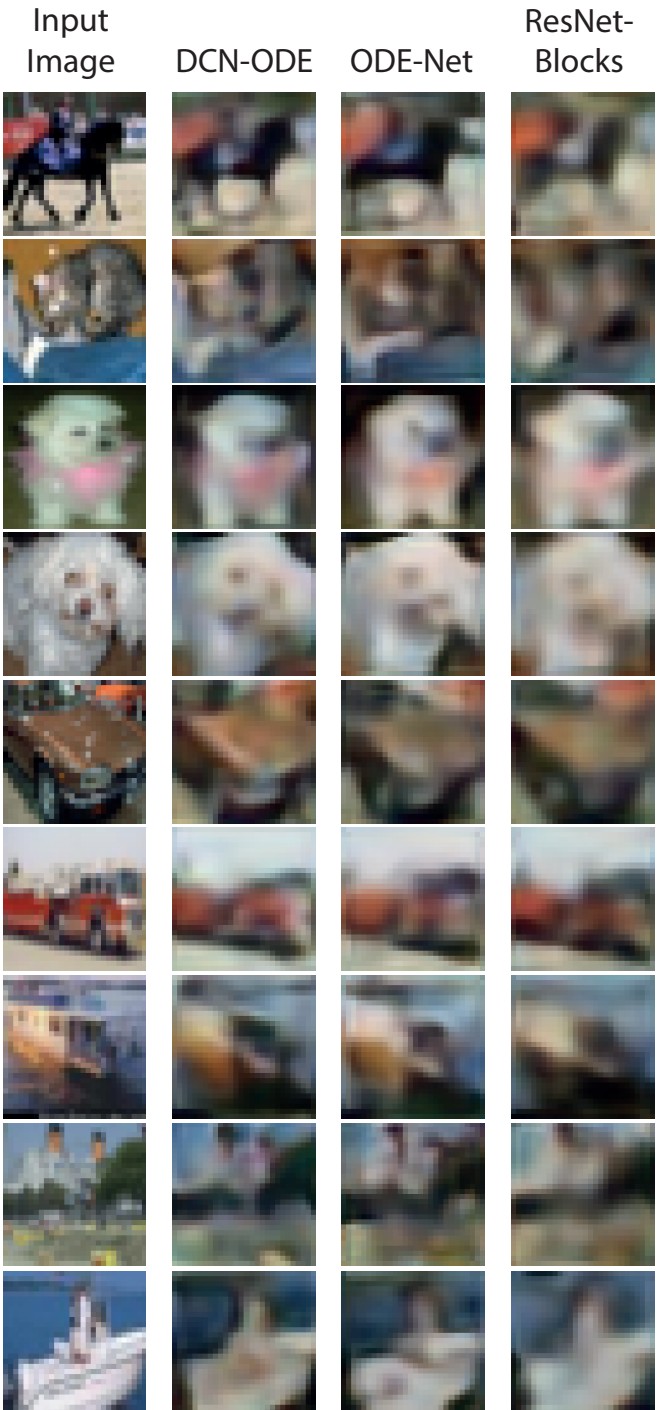

Figure 5: Example CIFAR-10 validation images and their reconstruction by the DCN-ODE model as compared to baseline models.

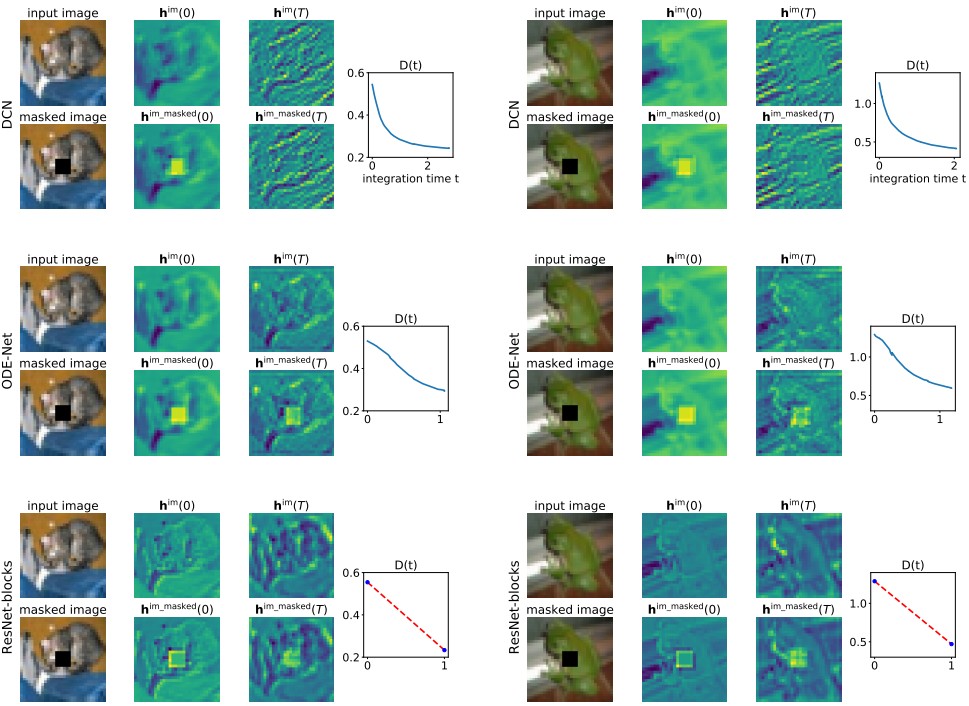

Figure 6: Feature map evolution within the first ODE block (or ResNet block) of different models.

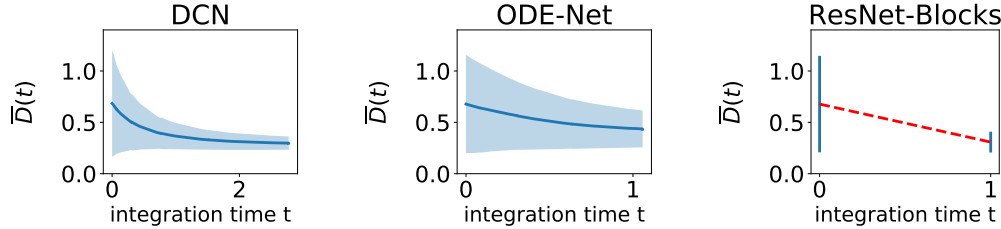

Figure 7: Evolution of the mean difference $D(t)$ between feature maps of an intact input image and a masked input image, averaged over 1000 images in the CIFAR-10 validation set. The shaded areas (or in the case of ResNet, the errorbars) show the standard deviation.

