# OpenReview forum: "Deep Continuous Networks"
_ICLR.cc/2021/Conference — Reject_

### Official Review · AnonReviewer2 · 2020-10-29

**Rating:** 5
**Confidence:** 4

**Review:**

Summary:

The authors propose a hierarchical model of neural ODEs, which they fit to CIFAR10. They find performance on par with ResNets, and include qualitative analyses on the filters learned by the models, their ability to fill in occluded features, and robustness to contrast at test time.


Strengths:

The filter parameterization is interesting. I can imagine this improving sample efficiency in certain contexts — perhaps the authors should seek out those kinds of tasks to complement their CIFAR results?

The discussion does a nice job of explaining the issues that neural ODEs have when scaling to large image datasets.


Weaknesses:

You spend time discussing spatio-temporal receptive fields throughout. Why? Your models are applied to 2d images.

The authors are missing a huge literature on (a) recurrent convolutional networks, and (b) using these networks to simulate classical vs. extra classical receptive field effects.

ResNet-Blocks is the original ResNet? Maybe change "blocks" to a citation? Or V1/V2 depending on which implementation it is (unclear from the text).

There's essentially no difference between the performance of any of the models tested. Is it possible to scale to ImageNet? It is important to show that the proposed method does *something* different than the standard ResNet. The authors attempted to add some qualitative experiments towards this goal in Fig 4, but those results are not very convincing. I think to show filling-in you'd want to show reconstruction in RGB space.

---

> ### Author Response · Authors · 2020-11-24
> **New results and subsections based on the reviewer's comments**
>
> We thank the reviewer for the positive comments. To address the suggestions and weaknesses:
>
> > The filter parameterization is interesting. **I can imagine this improving sample efficiency** in certain contexts — perhaps the authors should seek out those kinds of tasks to complement their CIFAR results?
>
> We have now added 2 extra experiments to better demonstrate the strengths of the model and performance improvements over baselines:
>
> 1. A 'restricted CIFAR-10' experiment, where we leverage the parameter efficiency of our proposed DCN models to show they perform **better than baselines in the small-data regime** (Table 2a).
>
> 2.  An image reconstruction experiment where DCNs achieve **lower loss than baselines** (Table 2b, Appendix A.4 and Fig. 5).
>
> > You spend time discussing spatio-temporal receptive fields throughout. Why? Your models are applied to 2d images.
>
> A big part of the motivation of our paper is to be able to accommodate more **biological complexity** than existing CNNs. We believe the proposed DCNs can be used to model the temporal profiles of neuronal responses, which are known to not be constant even when the experimental stimuli are static images: for example spatial frequency responses change over time in response to stationary gratings (Frazor et al., 2004). Similar observations are made for the contrast response function (Albrecht et al., 2002). Such temporal profiles cannot be simulated in conventional CNNs. We elaborate on this in Section 2.1.
>
> Temporally extending CNN dynamics also make DCNs more versatile as potential **models of biological circuits** in terms of physiological realism. For example, the in-built smooth evolution of filters (as in our meta-parametrized models) can be used to incorporate response dynamics such as synaptic depression or short-term potentiation. Likewise, the equations of motion can be modified to reflect axonal delays or generate oscillations. This is discussed in section 5.
>
> > The authors are missing a huge literature on (a) recurrent convolutional networks, and (b) using these networks to simulate classical vs. extra classical receptive field effects.
>
> We have now added a new subsection to related work (Section 4) to address these oversights. We thank the reviewer for this comment.
>
> > ResNet-Blocks is the original ResNet? (unclear from the text).
>
> We are sorry about the confusion, we now clarify this in section 2.4 as well as in the caption of Table 1.
>
> > There's essentially **no difference between the performance of any of the models tested**. It is important to show that the proposed method does **something different** than the standard ResNet. … you'd want to show reconstruction in RGB space.
>
> As detailed above, we have now added 2 extra experiments which demonstrate **performance improvements in DCNs over baselines**, including a data efficiency (Table 2a) and an image reconstruction (Table 2b, Appendix A.4 and Fig. 5) experiment as suggested by the reviewer.
>
> However, we also present the experiments related to pattern completion in feature space, as it serves a different purpose: Unlike reconstruction—where the networks are explicitly trained to reconstruct an image—pattern completion in feature space is a high-level phenomenon which emerges from the definition of DCNs. This is in-line with our motivations to propose a link between modern CNNs and well-known computational models such as the Ermentrout-Cowan model (Bressloff et al., 2001), and neural field models (Amari, 1977), which display interesting high-level phenomena (Coombes, 2005).

---

> > ### Comment · AnonReviewer2 · 2020-11-24
> > **Response**
> >
> > The new table 2 is very nice. I will increase my score.
> >
> > "A big part of the motivation of our paper is to be able to accommodate more biological complexity than existing CNNs..." I disagree with your framing and I do not think that your section on recurrent networks is adequate.
> >
> > (1) How does your approach "accommodate more biological complexity" than a ResNet? ResNets are universal function approximators. I do not understand this argument.
> >
> > (2) You are sticking with this "spatiotemporal" argument. I think if you did a more thorough dive into recurrent vision models, you'd see that what you are referring to are recurrent interactions between units. "Spatiotemporal" usually refers to the nature of the input (is it a movie or an image) [1]. I realize this is semantics, but the field has a standard and I do not think that you are consistent with that standard.
> >
> > (3) There is extensive work with recurrent vision models to incorporate constraints from biological circuits, including recurrent non-linearities you've mentioned. RNNs are also the standard approach for incorporating synaptic potentiation mechanisms [2, 3]. Perhaps focus more on what continuous-time dynamics could buy you over discrete-time processing of an RNN or (weight-tied) ResNet? That said, there is a wide gap between what is explored in this paper and biological models of vision, so keep this level of speculation in the Discussion.
> >
> > [1] Adelson T & Bergen J. 1985. Spatiotemporal energy models for the perception of motion.
> > [2] Ba et al., 2016. Using Fast Weights to Attend to the Recent Past.
> > [3] Miconi et al., 2018. Differentiable Plasticity: A New Method for Learning to Learn.

---

> > > ### Author Response · Authors · 2020-11-24
> > > **Thank you for the extra clarifications**
> > >
> > > We thank the reviewer for the positive comments. To address some of the concerns:
> > >
> > > (1) We apologize for the confusion, we did not clarify what we meant by 'biological complexity' very well. We think we understand your concerns better now. We did not mean to imply computational complexity by this: In the context of models in computational neuroscience, 'biological complexity' is sometimes used to refer to 'complex physiological and anatomical details' which are not captured by the model. For example a rate-based, coarse-grained model may not 'accommodate the biological complexity' of spiking activity, synaptic dynamics and conductances, variability in dendritic integration and so on. We believe by proposing an explicit model of learning the receptive field size in CNNs, we are able to capture more "biological realism".
> > >
> > > We have now fixed the text of the paper to read "biological realism" or "physiological realism" instead of biological complexity. Thank you for your input.
> > >
> > > (2) We understand the clash of terminology between different fields and agree with your assessment. We try to clarify this in the last paragraph of Section 4.
> > >
> > > To elaborate: Our use of 'spatio-temporal' is based on the use of the term in general in the field of non-linear dynamics (which DCN models derive from with the ODE definition) and based on the more specific use of 'spatio-temporal dynamics or patterns' generated by neurons in neuroscientific models (e.g. Coombes et al., 2005 from the manuscript; Walgraef, 2012, Spatio-temporal pattern formation: with examples from physics, chemistry, and materials science; Bressloff, 2011, Spatiotemporal dynamics of continuum neural fields; Roxin, Brunel and Hansel, 2005, Role of delays in shaping spatiotemporal dynamics of neuronal activity in large networks).
> > >
> > > However, we agree that it is not trivial to disentangle the time-variant dynamics from time-variant input. We tried to clarify this further in the last paragraph of Section 4.
> > >
> > > > I do not think that your section on recurrent networks is adequate.
> > >
> > > > (3) There is extensive work with recurrent vision models to incorporate constraints from biological circuits, including recurrent non-linearities you've mentioned. RNNs are also the standard approach for incorporating synaptic potentiation mechanisms [2, 3].
> > >
> > > We have now added the citations mentioned by the reviewer, as a part of the new subsection in Section 4. We would be happy to add (now or for the camera-ready version) any other relevant previous work the reviewers may recommend which we might have overlooked.
> > >
> > > > Perhaps focus more on what continuous-time dynamics could buy you over discrete-time processing of an RNN or (weight-tied) ResNet?
> > >
> > > Thank you, we also believe this is an advantage of the DCN model. In terms of the continuous-time dynamics captured by the neural ODE definition, we believe the modulation of the time-scale of the solutions with input scaling which we present in Figure 4b is an interesting concept, with computational efficiency benefits. This highlights the strength of the ODE framework, where the number of function evaluations is not fixed before training, over discrete-time networks.
> > >
> > > We now mention this in the second subsection of related work (Section 4).
> > >
> > > > That said, there is a wide gap between what is explored in this paper and biological models of vision, so keep this level of speculation in the Discussion.
> > >
> > > We now emphasize the gap between DCNs and biological models explicitly in the discussion, as well as cite the papers (Ba et al., 2016; and Miconi et al., 2018) in context in the discussion.

---

### Official Review · AnonReviewer3 · 2020-10-30
**Review DCN**

**Rating:** 7
**Confidence:** 4

**Review:**

## Review


### Summary

The authors define continuous deep networks by expressing 2D-convolutional filters as a linear combination of Gaussian function and its derivatives. By combining this description with the previously proposed neural ODE framework they obtain a spatio-temporally continuous description of a convolutional neural network.

There are 3 main contributions:

1. they are able to estimate the support width of the filters and they show that  it increases with the network depth as observed in the visual cortex
2. they show that their network performs as well as alternative non-continuous neural networks on CIFAR-10 while having less parameters
3. they exploit the temporal dynamics to resolve a pattern completion task

Overall, I think the work is good but I am not as enthusiastic as the authors about the importance of the work for neuroscience and machine learning.

### Strengths

* The paper is well written and easy to understand. The goals and contributions of the work are clearly stated.
* The quantitative results are marginally good.
* The qualitative results (filter support width, pattern completion and contrast robustness) are interesting and relevant to neuroscience


### Weaknesses

* I fail to understand how the work could be relevant to neuroscience beyond what is presented here. What is so important about using spatially continuous filters that couldn't be done with discrete filters ?
* The pattern completion task is not fully conducted. It would be great to reconstruct the missing part of the input.
* The increase of filter support width with network depth correlates with what is known for the visual cortex but I fail to understand how it could be relevant to current work in experimental neuroscience. What is the benefit of learning continuously changing support for machine learning ? About the relation to biology, the increase in size might be more related to the specific task on which the network is trained than to what is observed in the visual cortex.
* The observed contrast robustness is not compared to other neural networks nor discussed in the light of experimental neuroscience observations.

### Minor comments

* The text in the figures is way too small. It should be the same size as the main text.

---

> ### Author Response · Authors · 2020-11-24
> **New results and subsections based on the reviewer's comments**
>
> We thank the reviewer for the positive comments. To address the weaknesses:
>
> > The quantitative results are marginally good.
>
> We have now added 2 extra experiments to better demonstrate the strengths of the model and performance improvements over baselines:
> 1) A 'restricted CIFAR-10' experiment, where we leverage the parameter efficiency of our proposed DCN models to show they perform **better than baselines in the small-data regime** (Table 2a).
> 2) an image reconstruction experiment where DCNs can achieve **lower loss than baselines** (Table 2b, Appendix A.4 and Fig. 5).
>
> > I fail to understand how the work could be relevant to neuroscience beyond what is presented here. What is so important about using spatially continuous filters that **couldn't be done with discrete filters**?
>
> 1) Discrete filters have discrete kernel sizes (in units of pixels) which means it is not trivial to learn the kernel size during optimization via gradient descent.
> 2) It is also not trivial to scale (or rotate) discrete kernels at test time without interpolation artifacts. Similarly, it is not clear how to adapt RF shapes or sizes at test time to different inputs without a continuous kernel definition to sample from.
>
> This severely limits the ability of conventional CNN models to test biological or neuroscientific hypotheses about the **emergence of different receptive field (RF) sizes** as a function of input statistics or anatomical constraints. In contrast, our continuous description of the kernels have a continuous scale parameter which can be **learned during training**.
>
> In general, **accommodating more biological complexity** than conventional CNNs is one of the most important motivations for DCNs. Some relevant biological observations, which may not be trivial to model via discrete CNNs include (as mentioned in Section 2.1):
>
> - the change of RF size and spatial frequency selectivity as a function of input contrast (Sceniak et al., 2002),
> - the emergence of different RF sizes and contrast sensitivity as a function of different anatomical input connections (Bauer et al.,1999),
> - the change of the contrast response (Albrecht et al., 2002) and spatial frequency selectivity (Frazor et al., 2004), as a function of time.
>
> In this paper we do not focus on a single biological hypothesis, but on showing that DCNs can be successfully implemented with some generic parametrization (such as linear and quadratic meta-parametrization of temporal profiles). For future work, we believe our model presents a suitable framework for testing interesting biological hypotheses, similar to the work by Lindsey et al., 2019, but with the possibility of involving RF sizes and spatial frequency responses as learnable parameters.
>
> > The increase of filter support width with network depth correlates with what is known for the visual cortex but I fail to understand how it could be relevant to current work in experimental neuroscience.
>
> We do not explicitly increase the kernel size, but learn it, which is **not possible in conventional CNNs**. Being able to change kernel supports via a continuous scale parameter in CNNs **makes it possible to model** the biological observations described above (Sceniak et al., 2002; Frazor et al., 2004). In addition, the in-built smooth evolution of filters can be used, for example, to incorporate response dynamics such as synaptic depression or short-term potentiation. Likewise, the equations of motion can be modified to reflect axonal delays or generate oscillations. We now elaborate on these points in Section 2.1 and the discussion.
>
> > What is the benefit of learning continuously changing support for machine learning?
>
> Often deeper networks are used with increasing resolution of input images. This is done to achieve RF sizes which will be larger than the input image at the output level. However, such networks might be over-parameterized for simpler problems and problems in the small data regime. Learning the RF size, or increasing them with layers as necessary, can **relax the rather arbitrary dependence of network depth on input resolution**. The parameter reduction and data efficiency demonstrated in DCNs indicate learning a changing support width might be beneficial.
>
> > About the relation to biology, the increase in size might be more related to the specific task on which the network is trained than to what is observed in the visual cortex.
>
> In this paper, we choose CIFAR-10 as the training dataset because it contains relatively natural images compared to other datasets of small images (such as MNIST or Fashion-MNIST). However, we agree that this is an interesting hypothesis which remains to be tested, as the learnable scale parameter in DCNs makes it possible to test this hypothesis, and compare the behaviour and emergence of receptive field sizes to biological observations, which is not possible when using conventional CNNs.

---

> > ### Author Response · Authors · 2020-11-24
> > **Further comments**
> >
> > > The pattern completion task is not fully conducted. It would be great to reconstruct the missing part of the input.
> >
> > We have now included an RGB reconstruction experiment where DCNs outperform baselines (Table 2b, Figure 5). However, we also present the experiments related to pattern completion in feature space, as it serves a different purpose: Unlike reconstruction—where the networks are explicitly trained to reconstruct an image—pattern completion in feature space is a high-level phenomenon which emerges from the definition of DCNs. This is in-line with our motivations to propose a link between modern CNNs and well-known computational models such as the Ermentrout-Cowan model (Bressloff et al., 2001), and neural field models (Amari, 1977), which display interesting high-level phenomena (Coombes, 2005).
> >
> > > The observed contrast robustness is not compared to other neural networks nor discussed in the light of experimental neuroscience observations.
> >
> > We now expand upon these points in the beginning of Section 3.3.
> >
> > > The text in the figures is way too small. It should be the same size as the main text.
> >
> > We now increased the font size in every figure.

---

### Official Review · AnonReviewer4 · 2020-10-31
**Nice bridge between deep learning and computational neuroscience.**

**Rating:** 6
**Confidence:** 3

**Review:**

Summarize what the paper claims to contribute.
The paper develops a spatio-temporal network that is defined in terms of continuous spatial functions and continuous temporal dynamics. The approach is meant to bring deep networks closer to biological neural models. The model is tested on CIFAR-10, and on a variation of CIFAR-10 in which blocks of pixels are blacked out. The methodological novelty consists of combining several existing approaches (continuous kernels, kernel-scale learning, and neural ODEs).

List strong and weak points of the paper.
Strong points:
-	I think the motivation is very strong. Deep convolutional networks are increasingly used in brain modelling, but they are somewhat disconnected from earlier computational neuroscience in ways that this paper tries to address.
-	The pattern completion test with blacked-out pixels is a nice way to employ spatiotemporal dynamics in image recognition, and the results are promising.
-	It is very interesting that the distribution of learned scales reflects the distribution of receptive-field sizes in primary visual cortex.

Weak points:
-	The feature-map evolution results in Fig. 4 are interesting, and a few more examples are given in the appendix, but it would also be nice to see mean +/- SD dynamics across many images, to complement Figure 3C more thoroughly.
-	The model doesn’t outperform controls on CIFAR-10, although it does perform moderately better with blacked-out blocks of 6x6 pixels or more.
-	Performance of the model and baselines on CIFAR-10 is not strong, which raises the question of how compatible the approach is with higher performance.
-	The choice of CIFAR-10 as a test of the approach does not seem to be well motivated. A task with a temporal component might give the dynamic parts of the model more to do.

Clearly state your recommendation (accept or reject) with one or two key reasons for this choice.
I recommend to accept the paper. There is a growing body of work that compares deep CNNs to biological neural networks, and the conventions of discrete time and space in CNNs unfortunately distance this work somewhat from much previous computational neuroscience. I think the paper helps to close this gap.

Ask questions you would like answered by the authors to help you clarify your understanding of the paper and provide the additional evidence you need to be confident in your assessment.
-	Could you please expand on the motivation for continuous space with respect to biological vision? Ultimately vision is based on discrete photoreceptors, and generally on populations of discrete cells. Relatedly, the motivation for continuous time is more obvious, but I think it would also help to comment on spikes in this context.
-	The dynamic equation in Eq. 2 seems to be autonomous. How does input affect h?
-	In Section 3.2, the learned scales are shown to increase with network depth. This is compared with biological receptive fields, which grow through the visual hierarchy. But I had understood the scale to correspond to the kernel size rather than the receptive field size (which grows in deep networks even if all the kernels are the same size). Does sigma correspond to kernel or receptive field size?

Provide additional feedback with the aim to improve the paper.
-	Analytic tractability is mentioned on page 1 as an advantage of some continuous models in neuroscience, and it seems to be implied at that point that such benefits are sought in the paper, but it doesn’t seem that the approach ultimately offers much hope in this sense. If there is some potential here, please expand.
-	I didn’t understand “… time (or network depth)” on pg. 4.
-	Also on pg. 4, the sampling domain of the filter is given, but not the sampling frequency.
-	I found section A.1 relatively hard to follow. In particular, Eq. 5 seems to be meant to motivate the filter family, but I didn’t follow the argument (or maybe I missed the point entirely). Also, aside from general interest I didn’t understand how the paragraph that contains Eq. 6 related to the rest of the paper.
-	It wouldn’t hurt to define DOPRI.
-	I didn’t follow the last paragraph of A.2.

---

> ### Author Response · Authors · 2020-11-24
> **New results and subsections based on the reviewer's comments**
>
> We thank the reviewer for the positive comments. To address the stated weaknesses:
>
> >The feature-map evolution results in Fig. 4 are interesting, but it would also be nice to see **mean +/- SD dynamics across many images**, to complement Figure 3C more thoroughly.
>
> We have now added the mean and standard deviation of $D(t)$ for 1000 validation images in the main text (Figure 4a, bottom right) as well as in Appendix A5 (Figure 7).
>
> >The model **doesn’t outperform** controls on CIFAR-10.
> >Performance of the model and baselines on CIFAR-10 is not strong, which raises the question of how compatible the approach is with higher performance.
>
> We have now added 2 extra experiments to better demonstrate the strengths of the model and performance improvements over baselines:
> 1) A 'restricted CIFAR-10' experiment, where we leverage the parameter efficiency of our proposed DCN models to show they perform **better than baselines in the small-data regime** (Table 2a).
> 2) an image reconstruction experiment where DCNs can achieve **lower loss than baselines** (Table 2b, Appendix A4 and Fig. 5).
>
> >The choice of CIFAR-10 as a test of the approach does not seem to be well motivated. A task with a temporal component might give the dynamic parts of the model more to do.
>
> >The dynamic equation in Eq. 2 seems to be autonomous. How does input affect h?
>
> >I didn’t understand “… time (or network depth)” on pg. 4.
>
> We apologize about the confusion, we now clarify these points in text, right after Eq.3 and in a new subsection at the end of Section 4. To give a little more information:
>
> The ODE given in Eq.2 is not an autonomous differential equation in the classical sense of the term in that it is an explicit function of time ($t$-terms parametrized by $\textbf{d}^m$). However, if we understand the question correctly, you are right that **the network does not admit time-variant inputs** (such as a video). The input image only defines the initial conditions $\textbf{h}(0)$. In that sense, the neural ODE formulation that we adopt is different from modern recurrent networks (which can process time-variant input) and is more similar to feed-forward CNNs. In our descriptions, we tried to accommodate the readers who might be familiar with the neural ODE paper (Chen et al., 2018), where the number of function evaluations by the ODE solver was interpreted as 'network depth'.
>
> However, the proposed temporal extension of conventional CNNs is in-line with our motivation of providing a closer link between CNNs and models from computational neuroscience. Neuronal responses are not constant, but display temporal profiles, also in response to static stimuli (e.g. Albrecht et al., 2002; Frazor et al., 2004). Similarly, our temporally extended framework provides interesting possibilities  such as explicitly incorporating synaptic depression or oscillations, which we discuss in Section 5.
>
> > Could you please expand on the **motivation for continuous space** with respect to biological vision? Relatedly, the **motivation for continuous time** is more obvious, but I think it would also help to comment on spikes in this context.
>
> We now elaborate on this in a new paragraph at the end of Section 2.1. To quote the text:
>
> Ultimately, we see continuous representations in end-to-end trainable networks as a link between modern CNN architectures and computational models of biological vision. Although for small spatio-temporal scales it may be more appropriate to use discrete descriptions of biological neurons, such as populations located in spatially discrete locations, or to consider temporally distinct spiking dynamics, computational models using continuous population activity or rate-based models provide reasonably good explanations of **phenomena observed at the network or systems level**.  We believe such larger scale computational models align well with the purposes of computer vision.
>
> > I had understood the scale to correspond to the kernel size rather than the receptive field size (which grows in deep networks even if all the kernels are the same size). **Does sigma correspond to kernel or receptive field size?**
>
> We are sorry about the confusion. sigma corresponds to the kernel size and, as mentioned by the reviewer, the RF size typically also grows over layers in conventional convolutional networks. However, with fixed, constant kernel sizes, the RF size grows **linearly** as a function of depth (in the absence of downsampling). This is a limitation that the visual system does not necessarily have. It is not trivial to draw direct comparisons between different convolutional layers and different visual areas (although there is growing work to relate CNNs to the ventral stream, e.g. www.brain-score.org), but the growth of the RF sizes via pooling of inputs in the visual hierarchy does not necessarily seem to be linear (as in the meta-analysis in Fig. 9 of Smith et. al., 2001). We now clarified this in text, in the first paragraph of section 3.2.

---

> > ### Author Response · Authors · 2020-11-24
> > **Responses to additional feedback**
> >
> > > "Analytic tractability is mentioned on page 1 as an advantage of some continuous models in neuroscience and it seems to be implied at that point that such benefits are sought in the paper, but it doesn’t seem that the approach ultimately offers much hope in this sense. If there is some potential here, please expand."
> >
> > We now elaborate on this further in the discussion (Section 5). We did not pursue an analytical understanding of the DCN models within the scope of this paper. However, we believe both the ODE definition and our generally well-behaved definition of filters based on Gaussian derivatives allow for further analytical investigations (bringing existing knowledge from other fields including computational neuroscience and dynamical systems) as a benefit over conventional CNNs. Such investigations can also be beneficial for the computer vision community, for example, a better understanding of the time scale/input contrast trade-off (Fig. 4b) can be sought for computational efficiency.
> >
> > > Also on pg. 4, the sampling domain of the filter is given, but not the sampling frequency.
> >
> > We now fixed this (section 2.4).
> >
> > > I found section A.1 relatively hard to follow.  In particular, Eq. 5 seems to be meant to motivate the filter family, but I didn’t follow the argument (or maybe I missed the point entirely). Also, aside from general interest I didn’t understand how the paragraph that contains Eq. 6 related to the rest of the paper.
> >
> > We have now expanded the explanation in A.1 before and following Equation 5. We also elaborated on the motivations for our choice of the N-jet basis functions in A.1, including in the paragraph containing Eq 6.
> >
> > > It wouldn’t hurt to define DOPRI.
> >
> > > I didn’t follow the last paragraph of A.2.
> >
> > We now define the DOPRI algorithm in A.2. Similarly, we have expanded upon the last paragraph of A.2, where we explicitly mention how the DOPRI algorithm treats the time interval boundaries.

---

### Public Comment · ~Juntang_Zhuang1 · 2020-11-13
**Missing reference to a related work**

Hi, thanks for the nice work, I think the spatio-temporal view is very inspiring. I want to refer to our ICML paper, which provides a new method to more accurately estimate gradient in Neural ODE compared to the adjoint method. Since one of your contributions is the improved performance in standard classification task on Cifar, and we also achieved a high accuracy on Cifar10 (~95%) with Neural ODE, hence we would appreciate it if you could briefly discuss. (We use a larger model with Neural ODE, and I think your method could achieve similar accuracy with large models). Thanks much in advance.

[1] Zhuang, Juntang, et al. "Adaptive Checkpoint Adjoint Method for Gradient Estimation in Neural ODE." arXiv preprint arXiv:2006.02493 (2020).

---

> ### Author Response · Authors · 2020-11-24
> **Thanks for the pointer**
>
> Hi Juntang! Thanks for the feedback and your nice comment. We also find the performance increase provided by your ACA method quite impressive. It is now mentioned in our manuscript. Thanks again!

---

### Decision · Program_Chairs · 2021-01-07
**Final Decision**

**Decision:**

Reject

**Comment:**

This paper received 1 weak accept, 1 accept, and 1 weak reject.

All reviewers questioned the motivation for continuous space/time with respect to biological vision. Obviously, discrete approximations used in machine vision are approximations but it is not clear from the paper or the authors’ response that this severely limits the ability of deep nets to predict neural data in ways that their continuous nets would not.

In addition, I have to confess that I did not really understand the argument made by the authors in their revision. In any case, the burden should be on the authors to go beyond general statements and to really demonstrate that the proposed models provide actual insights for neuroscience since the performance in terms of machine vision on CIFAR10 is underwhelming (the authors have to find a low data regime and even then the reviewers stated that the baselines used are not strong baselines, the reduction in the number of parameters is quite small relative to methods for actually reducing the number of parameters).


Clearly, the work has potential as noted by the reviewers. The authors suggest that “DCNs can be used to model the temporal profiles of neuronal responses, which are known to not be constant even when the experimental stimuli are static images: for example, spatial frequency responses change over time in response to stationary gratings (Frazor et al., 2004). Similar observations are made for the contrast response function (Albrecht et al., 2002). Such temporal profiles cannot be simulated in conventional CNNs. “ This sounds like an interesting set of neuroscience data that the authors could be indeed leveraging to demonstrate the benefit of their approach. My recommendation would be to add those in a revision of this paper which will significantly strengthen the work. I would add that the concepts of temporal and spatial continuity are independent and the authors should consider studying them separately to provide more in-depth analyses and convincing results.

As it stands, the paper has clear potential but it does not make a sufficient contribution to either ML or neuroscience and hence, I recommend the paper to be rejected.